# A Systematic Review of Self-Report Instruments for the Measurement of Anxiety in Hospitalized Children with Cancer

**DOI:** 10.3390/ijerph18041911

**Published:** 2021-02-16

**Authors:** Gomolemo Mahakwe, Ensa Johnson, Katarina Karlsson, Stefan Nilsson

**Affiliations:** 1Centre for Augmentative and Alternative Communication, University of Pretoria, Private Bag X20, Hatfield 0028, South Africa; gomza.more@gmail.com (G.M.); ensa.johnson@up.ac.za (E.J.); 2Department of Health Sciences, Faculty of Caring Science, Work Life and Social Welfare, University of Borås, 501 90 Borås, Sweden; katarina.karlsson@hb.se; 3Institute of Health and Care Sciences, Centre for Person-Centred Care, Sahlgrenska Academy, University of Gothenburg, Box 457, 405 30 Gothenburg, Sweden

**Keywords:** anxiety, cancer, hospital, instrument, pediatric patient, pictorial support, self-report, symptom management

## Abstract

Anxiety has been identified as one of the most severe and long-lasting symptoms experienced by hospitalized children with cancer. Self-reports are especially important for documenting emotional and abstract concepts, such as anxiety. Children may not always be able to communicate their symptoms due to language difficulties, a lack of developmental language skills, or the severity of their illness. Instruments with sufficient psychometric quality and pictorial support may address this communication challenge. The purpose of this review was to systematically search the published literature and identify validated and reliable self-report instruments available for children aged 5–18 years to use in the assessment of their anxiety to ensure they receive appropriate anxiety-relief intervention in hospital. What validated self-report instruments can children with cancer use to self-report anxiety in the hospital setting? Which of these instruments offer pictorial support? Eight instruments were identified, but most of the instruments lacked pictorial support. The Visual Analogue Scale (VAS) and Pediatric Quality of Life (PedsQL™) 3.0 Brain Tumor Module and Cancer Module proved to be useful in hospitalized children with cancer, as they provide pictorial support. It is recommended that faces or symbols be used along with the VAS, as pictures are easily understood by younger children. Future studies could include the adaptation of existing instruments in digital e-health tools.

## 1. Introduction

Anxiety has been identified as a severe and long-lasting symptom experienced by hospitalized children with cancer [1]. The stress of hospitalization, healthcare professionals’ limited understanding of the children’s illness, limited coping strategies, and the pain of invasive medical procedures and treatment regimens have been found to be primary causes of anxiety in children with cancer [2]. Symptoms caused by multimodal treatment may also interfere with cancer treatment when patients discontinue their treatment due to their inability to cope with the symptoms [3]. Severe anxiety may also cause delays in treatment procedures, increase susceptibility to infection, and prolong recovery, thus affecting the overall cancer treatment and prognosis, while also decreasing patient satisfaction [2].

Unrecognized symptoms in children with cancer cannot be treated and may result in more intense subsequent symptom experiences [1]. Untreated symptoms also add further suffering and may result in significant psychosocial symptoms in children receiving cancer treatments [4]. For example, anxiety has been found to exacerbate pain, which, in turn, increases the children’s experience of anxiety and increases the need for sedatives [2]. Additionally, anxiety has been found to be an important predictor of the quality of life in children with cancer and, thus, if left untreated, may result in a decreased quality of life [1].

The worldwide incidence of cancer in children aged 0–19 years was found from 2001–2010 as 155.8 per million persons per year [5]. In most countries, improvements in cure rates have caused a shift of attention in the direction of addressing the psychosocial outcomes of cancer in children [4]. Psychosocial support is an important aspect for relieving emotional distress in children with cancer, as emotional disease is recognized as the sixth most vital sign in cancer care [6]. Screening, routine monitoring, and treating the symptoms of emotional distress should be conducted as regularly as for other vital signs in cancer care to provide psychosocial support for children with cancer [6]. It is, thus, important to assess and identify anxiety symptoms in children with cancer to provide effective symptom relief and adequate psychosocial care [4].

Healthcare providers may find it challenging to obtain self-reports from children, as children may not always communicate their symptoms due to their developmental level and corresponding age-related expressive and comprehensive language skills to give accurate responses to self-report measurement instruments or tools [7]. Moreover, younger children may not have acquired the suitable vocabulary to describe their symptoms, since this is dependent on their developmental and cognitive levels [8,9]. Therefore, when children have communication challenges, healthcare providers have resorted to acquiring the required information from their parents’ or caregivers’ proxy reports [10]. Proxy reports from parents have been reported as commonly used for children younger than eight years, as they were assumed to lack the cognitive ability to provide accurate symptom descriptions and may not be a true reflection of the child’s symptom experiences—especially for psychological symptoms, such as anxiety [11]. However, healthcare providers have also been found to under-report the frequency and severity of treatment-related symptoms compared to child self-reports [12].

Self-reports are especially important for emotional and abstract concepts, such as anxiety [13]. Thus, self-reports should be used in the assessment of anxiety in children, as children are the best sources of information about themselves [14]. According to Kestler and Lobiondo-Wood [15], by the age of five years old, children are capable of communicating relevant and consistent information regarding their symptom experiences provided that appropriate instruments are used. Vatne et al. [9] emphasized that children should voice their symptoms and be involved in the planning of interventions, as this is their human right.

Self-report instruments require verbal and/or language abilities [7,8]. Self-reports of anxiety in children with cancer pose a challenge to healthcare providers due to these children’s inability to communicate, which is often influenced by language barriers, developmental levels, cognitive abilities, and degree of illness. Communication is key to obtaining self-reports of anxiety, as performance tests, such as cortisol and adrenaline measures, may present to be difficult, and, as such, verbal instruments are required for these assessments [16]. A common language is another required component for effective communication with children to ensure they accurately report their symptoms during an assessment. Effective communication occurs when a healthcare provider understands and integrates information gathered from the patient, and the patient in turn comprehends the healthcare provider’s message in a manner that permits active and responsible participation [10]. Verbal communication skills and a common language are, thus, essential for children to effectively communicate their symptoms.

In a systematic review by Lazor et al. [4], the included articles reported that the Visual Analogue Scale (VAS) was the most commonly used single-item instrument, and the State-Trait Anxiety Inventory (STAI) was the most commonly used multi-item instrument. According to Han [17], the STAI was a commonly used self-report instrument used in pediatric patients, although a more rigorous evaluation of the psychometric properties of the instruments and the cross-validation of different age groups are both lacking. The STAI also has no pictorial support for children with limited language, cognitive, or communication abilities.

Foster and Park [2] conducted an integrative study in which instruments, e.g., STAI for children (STAIC), were identified. Several limitations regarding the instruments were reported, such as a lack of clinical feasibility and being too long or too complex for the attention span of a sick child. For example, first, the STAIC was reported as being difficult for children to complete in a busy hospital setting. Secondly, additional research is required to establish the reliability and validity psychometric components of the instruments. Thirdly, all the instruments lacked validation across different levels of cognition, emotional and language development. Fourthly, the cultural appropriateness and sensitivity of the instruments have been questioned, as there is limited evidence of testing in diverse cultures, i.e., the instruments were only tested in white children in the United States [2].

Additional challenges experienced with symptom measurement instruments included the use of adapted adult version measurement tools, which were inappropriate for the children’s cognitive and developmental level. This included, for example, the Memorial Symptom Assessment Scale (MSAS), which was adapted for children aged 10 to 18 years [18]. The children given this scale had difficulty completing the test for various reasons, including visual impairments, poor comprehension of the test, and severe illness [18]. Due to the scarcity of age-appropriate measurement instruments or tools, methods to assess children’s symptoms have been identified as limited, which contributed to the poor management of the symptoms [8].

Healthcare providers need to offer validated and reliable self-report instruments or tools that can easily be administered to children (5–18 years) to assist them to effectively communicate their anxiety symptoms regardless of their communication, reading, and cognitive abilities. Specifically, valid, reliable, and age-appropriate measurement instruments that provide pictorial support are crucial to measure anxiety levels in children in order to better guide the selection of appropriate interventions. The purpose of this review was to systematically search the published literature and identify validated and reliable self-report instruments available for children aged 5–18 years to use in the assessment of their anxiety to ensure they receive appropriate anxiety-relief intervention in hospital.

The research questions were: What validated self-report instruments can children with cancer use to self-report anxiety in the hospital setting? Which of these instruments offer pictorial support?

## 2. Materials and Methods

### 2.1. Study Design

A systematic review methodology was conducted to comprehensively search, locate, and concisely synthesize all published studies in which validated measurement instruments were used to assess anxiety in children with cancer [19]. A systematic review is a type of literature review that entails a comprehensive search, analysis, and integration of evidence from multiple studies that have been evaluated using pre-defined eligibility criteria to answer the purpose of the study and the specific research questions [19,20]. The advantage of this method is that it allows for the reduction of potential bias through a process of critically appraising the quality of the methodologies used in the included studies [21]. A rigorous process is followed in terms of systematic reviews of the method of collection, appraisal, aggregation, and interpretation of relevant studies, which provides reliable findings to draw accurate conclusions; hence, these rigorous reviews are preferred over other review types [20]. This study was conducted in accordance with the Preferred Reporting Items for Systematic Reviews and Meta-Analyses (PRISMA) [20].

### 2.2. Search Strategy

The search terms were finalized through a scoping search in MEDLINE (Proquest) and CINAHL. A comprehensive search of the literature was conducted during February–March 2020, in the following electronic bibliographic databases: MEDLINE (Proquest), PubMed, PsycInfo, CINAHL, Scopus, and ERIC (Ebscohost and Proquest). The BOOLEAN search terms were cancer OR oncol* OR neoplasm AND anxiety AND paedi* OR pedi* OR child* OR adoles* OR school-age* OR kindergar* OR pre-school* AND assess* OR measure* OR scale* OR tool* OR eval* OR test* AND hospital OR med*.

### 2.3. Inclusion and Exclusion Criteria

All studies indexed and published until January 2020 were included in the search if they met the following criteria: The selected study’s population should include hospitalized children between 5 and 18 years of age with cancer. The study had to discuss the psychometric properties of self-report instruments for anxiety. As a score of 7 (50%) is regarded as good quality [22] on the Quality Assessment of Diagnostic Accuracy Studies (QUADAS) tool for methodological quality, only studies that scored 7 (50%) or higher were included in the systematic review. In addition, only studies published in English were included. The exclusion criteria included all types of reviews, editorials, protocols, case reports, short communication, and non-empirical (theoretical and discussion) papers.

Studies that met the criteria were identified at the title level by the first author. Once the titles were identified, an independent abstract review by the first two authors was conducted. A 92% agreement was reached between the two reviewers. Conflicts were discussed until 100% consensus was reached on which studies to include at the full-text level. The first two authors then obtained and reviewed full-text studies and conducted a quality appraisal of the methodology of these studies for final inclusion.

### 2.4. Quality Appraisal

Before data extraction was conducted with the full-text studies, a critical appraisal of the selected studies was conducted by the first two authors. The QUADAS tool was used to assess the methodological quality of the included studies [22]. The tool has a list of 14 questions to be answered with ‘yes’ (1), ‘no’ (0), or ‘unclear’ (0) to achieve a maximum score of 14 [22]. The items covered the patient spectrum, reference standard, disease progression bias, verification bias, review bias, clinical review bias, incorporation bias, test execution, study withdrawals, and indeterminate results [22]. Studies with a score less than 7 (50%) on the QUADAS tool were excluded.

### 2.5. Data Extraction

After the quality appraisal, the first two authors extracted the data from the studies that scored 50% or higher on a custom-made form to collect the relevant information to answer the purpose of the study and its research questions. Data extraction was checked and confirmed by the last two authors.

## 3. Results

### 3.1. Search Results

The search strategy identified 1269 possible studies, of which 1054 were excluded via title review. Thus, the remaining 215 met the inclusion criteria at the title level. After 76 duplicates were removed from the 215 studies, the remaining 139 studies were uploaded on Rayyan (http://rayyan.qcri.org) to be reviewed at an abstract level by the first two authors. Rayyan is a web and mobile application for systematic reviews developed to accelerate the process of the initial screening of abstracts and titles for inclusion and exclusion [23]. Figure 1 portrays the PRISMA flow diagram [20], indicating the number of studies identified at the title level, the selection of the number of the studies at the abstract and full-text levels, as well as their eligibility, and the final number of selected studies included in the review.

From the 139 study abstracts, 85 studies were excluded as they did not meet the inclusion criteria. A 92% agreement was reached, with a total of 11 conflicts (8%) identified at the abstract level. After discussion and mutual agreement between the two reviewers, these conflicts were resolved, and 100% consensus was reached on the studies considered eligible for full-text inclusion. As such, a total of 54 studies were identified to be reviewed at the full text level. With the exception of three studies that were not available, the full texts of 51 studies were retrieved and uploaded on Rayyan. To meet the study aims, it was important to ensure that the studies reported on the psychometric qualities of the self-reported instruments. Therefore, 25 studies were excluded on the full text level because it did not provide information on the validity of the instruments. A total of 26 studies were selected for methodological quality appraisal, and, after this quality review, 19 articles remained included in the study (Figure 1).

Table 1 outlines the results of the assessment of the critical appraisal scores. Overall, 19 out of the 26 included studies (73%) were categorized as methodologically high quality (score 7 to 14; 50% or above) and included in this study. The remaining seven (37%) studies were of a low quality (below score of 7; below 50%) and were, thus, excluded from the study.

### 3.2. Characteristics of the Selected Studies and Anxiety Instruments

Table 2 presents the characteristics of the selected studies and self-reported anxiety instruments. The characteristics of the studies include the authors, date, country where the study was conducted, aims, and the QUADAS scores of each study. The characteristics of the eight validated self-reported anxiety instruments include the report format of the instruments and if the instruments include pictorial support and the population with whom the instruments were validated with, as well as the psychometric qualities, for each instrument as reported within each study.

Table 3 provides a summary of the included 19 studies that were published between 1997 and 2020. Ten (53%) of the 19 studies were published after 2010. Sixty-five percent of the studies were conducted either in the USA (*n* = 3; 16%), Sweden (*n* = 3; 16%), Australia (*n* = 2; 11%), Canada (*n* = 2; 11%), or Taiwan (*n* = 2; 11%). The study population included participants with cancer between 1 and 39 years of age, albeit all studies included children with cancer within the range of 5 to 18 years of age. Although the selection inclusion criteria stated participants within the 5 to 18 age range, it was decided to include the Ray et al. study [24], since 25% of the participants were in the age group below 19 years. The McCarthy et al. study [25] included participants up to 25 years with a mean age of 21 years 6 months, indicating that the majority of the participants’ ages were closer to the required 18 years age criteria cut-off. Although some studies also reported on other assessment instruments, only self-reported anxiety instruments with reported psychometric properties were included in the review (see Table 2 and Table 3).

From Table 3, it is clear that eight self-report instruments were identified in the 19 studies (see Table 2 for a summary of these eight instruments). The instruments included one single-item instrument, the VAS [26], and seven multi-item instruments, namely the Hospital Anxiety and Depression Scale (HADS), the Kessler Psychological Distress (K10), the Patient-Reported Outcomes Measurement Information System (PROMIS), the Pediatric Quality of Life (PedsQL™) 3.0 Brain Tumor and PedsQL™ 3.0 Cancer Module, the Revised Child Manifest Anxiety Scale (RCMAS/RCMAS-2), and the State Trait Anxiety Inventory (STAI—including STAI-S, STAI-T, STAI-C). This review only focused on the self-report instruments.

In terms of the research question about pictorial support, three instruments were found to have pictorial support, namely the VAS and the PedsQL™ 3.0 Brain Tumor and PedsQL™ 3.0 Cancer Module, which included a faces scale for five- to seven-year-old children.

#### 3.2.1. Single-Item Instruments

##### Visual Analogue Scale (VAS)

The VAS is a self-report instrument that in this study assessed anxiety in children with cancer seven years of age and older. The instrument offers pictorial support in the form of a 100-mm horizontal line with anchors at the extreme ends. Respondents are required to indicate the level of their anxiety on the 100 mm horizontal line from 0 to 100, where 0 denotes no anxiety, while 100 denotes the worst possible anxiety [26].

#### 3.2.2. Multi-Item Instruments

##### Hospital Anxiety and Depression Scale—Anxiety (HADS-A)

The HADS is a 14-item self-report questionnaire to assess anxiety and depression in adults and adolescent patients; it was used in this study for 13- to 19-year-old adolescents [28]. Other age groups included in this systematic review were 15–19 years [24] and 12–17 years [30].

The HADS consists of two subscales, namely anxiety (HADS-A) and depression (HADS-D), each comprising seven items [28]. Since the focus of this review is only on anxiety instruments, information of only the HADS-A has been presented. The anxiety scale is scored using a four-point Likert scale ranging from 0 to 3, with higher scores indicating the presence of anxiety. The range of scores is 0 to 21, where scores of 0 to 7 signified no anxiety/depression, 8 to 10 denoted mild to moderate anxiety, and 11 to 21 denoted moderate to severe anxiety [29]. No pictorial support was described for this instrument.

##### Kessler Psychological Distress Scale (K10)

The K10 is a 10-item self-report questionnaire, which measures global distress in adolescents and young adults (AYA) with cancer between the ages of 15 to 25. The questionnaire asks respondents to rate their anxiety and depression symptoms, which occurred over the past four weeks, on a five-point scale. The score ranges from 10 to 50 [25]. The instrument offers no pictorial support.

##### Patient-Reported Outcomes Measurement Information System (PROMIS)

The English version of PROMIS [32] included the age group 7 to 18 years. The Chinese pediatric PROMIS (C-Ped-PROMIS) was used in this study with 8- to 17-year-olds [31]. The scale consists of two short forms of Anxiety and Depression. The C-Ped-PROMIS measures symptoms of anxiety and depression experienced in the preceding seven days on a five-point Likert scale from 0 to 4, where 0 = never, 1 = almost, 2 = sometimes, 3 = often, and 4 = always. The Anxiety short form has eight items. A score of 70 was considered a high score, and 80 was very high, while 30 was a low score, and 20 was a very low score. Higher scores meant the measured symptom was experienced more [31] but offers no pictorial support.

##### PedsQL™ 3.0 Brain Tumor Module

The PedsQL™ Brain Tumor Module is a multidimensional assessment instrument with both self-report and proxy report versions. The instrument consists of six scales: cognitive problems (seven items), pain and hurt (three items), movement and balance (three items), procedural anxiety (three items), nausea (five items), and worry (three items) [33]. The proxy report version is for toddlers aged two to four and does not include the Cognitive Problems Scale, while the child- and parent-reports for young children (aged five to seven) list only six items on the Cognitive Problems Scale. Respondents are asked to describe the extent to which each symptom has been bothersome to them over the past seven days. Responses are rated on a five-point Likert response scale, with scores indicating 0 = never (a problem); 1 = almost never; 2 = sometimes; 3 = often; and 4 = almost always on the child-reports for those ages 8 to 18 years and all parent-reports. The instrument offers pictorial support in the form of a three-point face response scale to aid participants aged five to seven years old in understanding the concept of rating scales and self-report their symptoms [33].

##### PedsQL™ 3.0. Cancer Module

The PedsQL™ 3.0 Cancer Module is a 27-item multidimensional instrument with self-reports in the age group 8–17 years old [34]. For children between the ages of five and seven years, there are only three response options: ‘never’, ‘sometimes’, and ‘almost always’. In addition, three pictures of facial expressions varying from a smiling face to a very sad face indicate no problem/no difficulty/no pain to a lot of problems/difficulty/worst pain [35]. The broad age span of the scale, 2–18 years (with by proxy versions), makes it possible to compare different age groups [36].

##### Revised Child Manifest Anxiety Scale (RCMAS/RCMAS-2)

The RCMAS is a 37-item self-report questionnaire, which assessed the level of anxiety in children and adolescents [38]. The instrument is a revised form of the Children’s Manifest Anxiety Scale (CMAS), with a yes or no response format. The RCMAS was in the studies included the age groups 6 to 17 years [38], as well as 10-16 years [39]. The questionnaire has no pictorial support.

The Revised Children’s Manifest Anxiety Scale second edition (RCMAS-2) contains 40 items to measure anxiety [40], and exists also in a short form with 10-items [37]. This study included 370 participants in the age group 6–19 years. This study supported the validation of the RCMAS-2 to measure anxiety in pediatric cancer patients [40]. The study with the short form, included the ages 8 to 19 years [37].

##### State Trait Anxiety Inventory (STAI) Trait and State Scale and State Trait Anxiety Inventory for Children (STAIC)

The Spielberger STAI is a 40-item, self-report anxiety questionnaire with two subscales of State and Trait anxiety. The scale was originally developed to measure anxiety in normal adults, although it is now used for the assessment of anxiety levels in both adolescents and adults aged 12–20 years [41]. The other study had the age group 8–17 years [34]. The State Trait Anxiety Inventory—State scale (STAI-S) is a 20-item self-report inventory [41]. The State Trait Anxiety Inventory—Trait scale (STAI-T) is a 20-item self-report inventory [42].

The STAIC is a self-report rating scale comprising two sections of 20-item state and trait anxiety for children between 8 and 18 years old. The STAIC is scored on a 3-point Likert scale rating from 1 to 3, with a score range from 20 to 60 and total scores for the two scales calculated separately. Higher scores depict higher levels of anxiety. The STAIC is one of the most frequently used instruments in pediatric research and in the assessment of anxiety in children, although no pictorial support is offered [33].

## 4. Discussion

Anxiety is a subjective symptom and should, thus, be reported by the individual child, as he or she best can describe the experience [11]. Anxiety is a psychosocial symptom and abstract concert, whereby self-reports of the symptom are better and more accurate than proxy reports [11,13]. Therefore, self-report anxiety instruments were a priority in this review; children should be offered the opportunity to self-report their anxiety symptoms, as they are the best sources of this information and will give a true indication of their symptom experience [15]. In doing so, they fulfill their human right to active participation in their own healthcare [43].

In this review, a total of eight self-reported anxiety instruments were identified of which three had pictorial support (100 mm VAS, PedsQL™ 3.0 Brain Tumor Module and PedsQL™ 3.0 Cancer Module). Instruments that offer pictorial support may be suitable for use by communication-vulnerable children, i.e., children with verbal communication difficulties and language barriers [44], to enable them to self-report their symptoms. Pictures have been proven to be more effective in describing subjective emotions, such as anxiety [1]. Thus, pictorial supports are typically preferred for obtaining self-reported anxiety from children with cancer or with communication disability as pictures are more easily understood by younger children. Apart from the 100 mm VAS, a single-item instrument, the other seven instruments were multi-item instruments.

The 100 mm VAS (single-item instrument) was administered only once in the present analysis and offers pictorial support. The 100 mm VAS is a valid and reliable self-report instrument for the assessment of anxiety levels in children. Besides, the instrument has been reported to be practical and easily comprehended by children seven years of age and older [26]. The instrument offers pictorial support in the form of a 100-mm or 10-cm horizontal or vertical line; however, it requires the children to be capable of reading numbers and understanding space and distance [2]. Therefore, the option to add a form of a faces scale (from a smiling face to a very sad face) to supplement the horizontal or vertical line could be investigated when designing assessment instruments for digital tools.

The HADS has both self-report and proxy report versions and demonstrated adequate test-retest reliability and sensitivity for use with adolescents [28]. Furthermore, the instrument has normative data in the Swedish population [45] and a Chinese version with good validity and reliability [29]; thus, it has some evidence of generalization in different populations. However, the instrument has no pictorial support. This proves that the instrument can be used for adolescents with cancer; however, it may not be suitable to measure self-reported anxiety in younger children with cancer.

The K10 is a questionnaire that has been used in adolescents and young adults in Australia [25]. The instrument is also translated in different languages, e.g., Arabic [46], Chinese [47], and Danish [48]. It has been used for both adolescents with cancer [25] and their siblings [47]. The K10 was used in a study to describe unmet needs and distress in bereaved offspring and bereaved siblings. The results showed that participants with greater levels of psychological distress also reported a higher number of unmet needs [49]. No pictorial support was mentioned in the present review.

The PROMIS instrument is also appropriate and valuable for the assessment of self-reported anxiety in children with cancer. The PROMIS instrument demonstrated good reliability and validity in children with cancer [32] and has some evidence of good cross-cultural validity [4]. This implies that the instrument may be adapted for different cultures and languages, thus overcoming significant language barriers. Likewise, the instrument can be administered using computerized-adaptive testing technology [32] and, thus, may be valuable in the design of a digital communication tool. The PROMIS instrument enables children to provide self-reports of their symptoms, although no pictorial support was mentioned in the present review, making it possibly unsuitable for the communication-vulnerable population.

The PedsQL™ 3.0 Brain Tumor Module was designed to measure brain tumors, specifically health-related quality of life with child self-report from 5 to 7, 8 to 12, and 13 to 18 years [50]. Internal consistency reliability was demonstrated for the 24-item PedsQL™ 3.0 Brain Tumor Module with a Cronbach alpha of 0.76–0.87 for child self-report [50]. Results from the current review confirm the reliability of specifically the procedural anxiety scale of the Japanese translated version of the PedsQL™ 3.0 Brain Tumor Module that falls within the original Cronbach alpha score [33]. Although only the self-report form for the 5- to 7-year-old children include pictorial support, it is suggested to investigate the use of pictorial support for older age groups in an attempt to assist communication-vulnerable children to also provide self-report. Once again, it should be remembered that the original purpose of the PedsQL^TM^ 3.0 Brain Tumor Module focused on health-related quality of life and not only anxiety [50].

The PedsQL™ 3.0 Cancer Module was specifically developed for the pediatric cancer population [51]. The cross-cultural validity and reliability of the instrument has been well-established in the Brazilian population [51]. A good internal reliability—with a Cronbach’s alpha value between 0.76 and 0.80—for the standardized scale was reported for the instrument [34,35]. Thus, the PedsQL™ 3.0 Cancer Module is valid and reliable for use by children with cancer aged two to 18 years old [34]. The adapted version of the instrument with pictorial support for children aged five to seven years [36] could provide the opportunity of using the PedsQL™ 3.0 Cancer Module with communication-vulnerable children. Another positive aspect is that the instrument is available in three separate versions for ages five to seven, eight to 12, and 13 to 18 [51]; thus, it allows communication-vulnerable children with cancer to self-report their symptoms. The PedsQL™ 3.0 Cancer Module is suitable, reliable, and valid and could be used in the design of digital communication tools, since it has the required pictorial support and can be used in the hospital setting to obtain self-reports of anxiety symptoms in children with cancer. However, the instrument specifically assesses the impact of anxiety on the quality of life of children with cancer [52] and not necessarily their anxiety levels.

The RCMAS-2 is another multi-item instrument that was identified in this review as having adequate psychometric properties of reliability and validity [40]. The RCMAS-2 has extensive cross-cultural evidence, including in African countries, such as Zimbabwe [40]. For example, the Portuguese version was evaluated to have a Cronbach’s alpha of 0.63 for the children’s self-report version [37], whereas the original scale had a Cronbach’s α of 0.92 [40]. No pictorial support was mentioned in the present review.

Another instrument found reliable and valid for use by children with cancer is the STAI/STAIC. The STAIC, a widely used self-report anxiety assessment instrument for children [53], has been used in many studies in the cancer population [54]. The original instrument has proven to be valid and reliable, with a Cronbach’s value of 0.81 [34]. Furthermore, the instrument had good validity and reliability for its Japanese [33], Spanish [53], and Turkish versions [54]. Thus, it has good cross-cultural validity and generalization. The STAIC is a valid and reliable instrument for the assessment of self-reported anxiety in children [52] and is suitable for use in children with cancer in a variety of settings, including the hospital setting [4]. Regrettably, no pictorial support is offered for the communication-vulnerable population. Foster and Park [2] also highlighted that the instrument uses terms not easily understood by children aged seven to 12 years, which might diminish its usability in children with cognitive deficits and receptive language delays.

Similar to the findings by Lazor et al. [4], the STAI was found in the present analysis to be one of the most commonly used multi-item instruments for children, although it does not offer pictorial support. As expected, all eight instruments (e.g., the VAS, HADS, K10, PROMIS, PedsQL™ 3.0 Brain Tumor Module, PedsQL™ 3.0 Cancer Module, RCMAS/RCMAS-2, and STAI/STAIC) were identified to be valid and reliable in the measurement of anxiety in children with cancer, which making them valid self-report instruments. The quality of assessment results, in general, depends on the selection of an instrument that provides valid and reliable measures; hence, instruments deemed valid and reliable were identified in this review. Systematic errors are reduced in valid and accurate instruments; therefore, healthcare providers are required to carefully select instruments with good psychometric properties [55].

In this review, three anxiety instruments offered pictorial support: the VAS (horizontal line to indicate intensity), PedsQL™ 3.0 Brain Tumor Module, and PedsQL™ 3.0 Cancer Module (offering faces scales for younger children). The pictorial support helps children understand what was asked by the instrument when they lack adequate vocabulary to communicate verbally [7,8]. This demonstrates that, in situations where children do not understand the questions of an anxiety instrument or may not understand their healthcare provider due to language barriers, alternative strategies, i.e., pictorial support, could play a crucial role in assisting them in communicating their anxiety. Moreover, healthcare providers have to be able to obtain self-reports of anxiety from these children in order to plan and provide appropriate anxiety-relief interventions.

Several strengths of this review can be noted. This study was conducted with methodological rigor through the conduction of a critical appraisal of the included studies (using the QUADAS tool), the following of the PRISMA guidelines, and the use of two independent reviewers in the selection of the included studies at both the abstract and full-text level. The psychometric properties of the identified anxiety measurement instruments were discussed to allow for the accurate selection of reliable and valid instruments for the assessment of anxiety in hospitalized children with cancer. Another strength of this study is that its data extraction was conducted by two independent reviewers, with a third reviewer who was consulted when no clear decisions could be made by the first two reviewers. A number of limitations, however, also applied to this study. First, only studies that were published in English were included. Second, some full-text articles were inaccessible. Due to these limitations, it is possible that some self-report tools for anxiety were missed. Last, the QUADAS tool is a valid tool for critical appraisals of systematic reviews. However, there is an element of subjectivity in this review process; thus, other reviewers in other studies may exhibit differences in their ratings and synthesis. Future studies should consider using three or more reviewers during the quality appraisal phase.

Although anxiety is among the most frequently experienced cancer symptoms in children, few instruments with pictorial support are available to help children communicate and self-report their symptoms. Severe anxiety may cause delays in treatment procedures, increased susceptibility to infection, and prolonged recovery, thus affecting the overall cancer treatment and prognosis and also resulting in decreased patient satisfaction [2]. Therefore, it is suggested for future studies to amend some existing anxiety instruments by providing pictorial support to enable children, especially communication-vulnerable children, to communicate and self-report their anxiety. Since there is an ongoing global shift toward digital tools, future studies could focus on the development of self-reported anxiety instruments used as part of e-health tools.

## 5. Conclusions

A comprehensive review of the literature revealed only eight instruments with good psychometric properties that were reliable and valid for use in the assessment of anxiety in hospitalized children with cancer. Most of the identified instruments, however, lacked pictorial support, which limits their use in children with communication challenges and limited or low literacy skills. The VAS, PedsQL™ 3.0 Brain Tumor Module, and PedsQL™ 3.0 Cancer Module especially proved to be useful in hospitalized children with cancer, as they provide pictorial support (i.e., faces scales). Furthermore, it is recommended that faces scales or symbols be used along with the VAS, as pictures are easily understood by younger children. These instruments may be adapted for digital tools.

## Figures and Tables

**Figure 1 ijerph-18-01911-f001:**
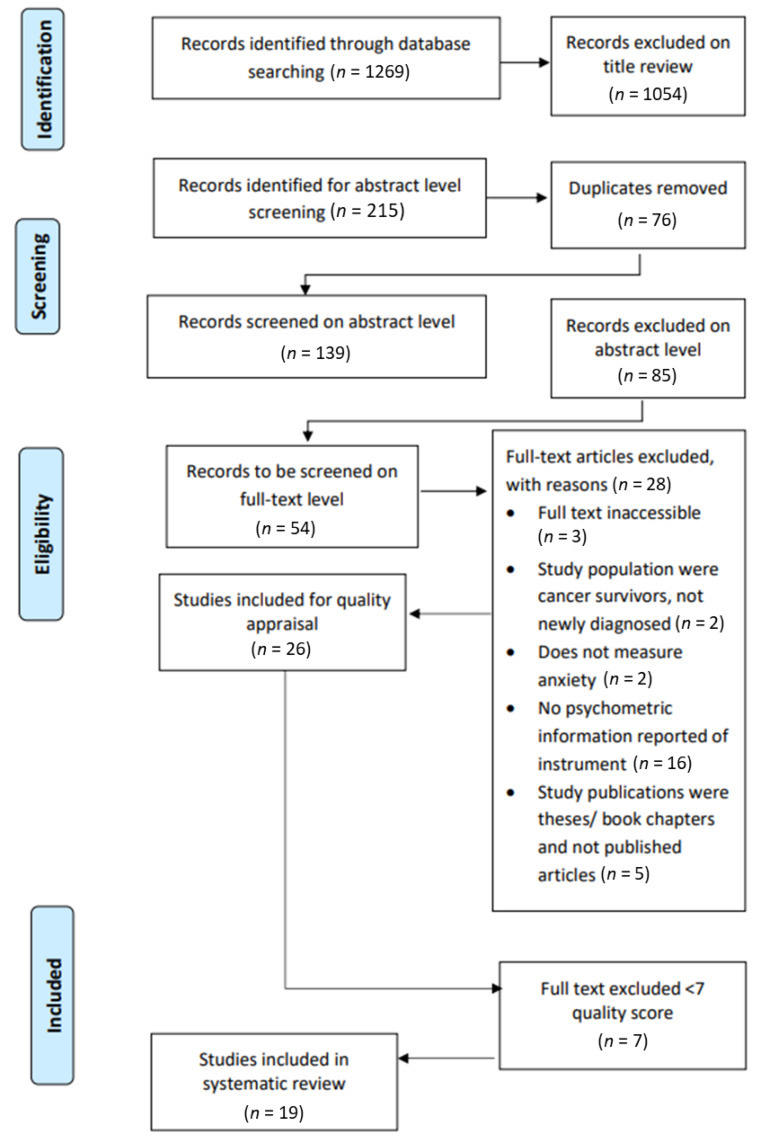
Preferred Reporting Items for Systematic Reviews and Meta-Analyses (PRISMA) flow diagram of the selection process.

**Table 1 ijerph-18-01911-t001:** Critical appraisal scores.

	1. Spectrum Criteria	2. Selection Criteria	3. Reference Standard	4. Disease Progression Bias	5. Partial Verification	6. Different Verification	7. Incorporation Bias	8. Index Test Execution	9. Reference Standard Execution	10. Test Review Bias	11. Reference Standard Review Bias	12. Clinical Review Bias	13. Uninterpretable Results	14. Withdrawals	Total
Allen et al., 1997	1	1	1	1	0	1	1	1	1	0	0	1	0	1	10
Burgess and Haaga, 1998	0	1	1	1	0	0	1	1	1	0	0	1	0	0	7
Germann et al., 2015	1	1	1	1	1	1	1	1	1	0	0	1	0	1	11
Hedström et al., 2005	0	1	1	1	1	1	1	1	1	0	0	0	1	1	10
Jörngården et al., 2007	0	1	1	1	0	1	1	1	1	0	0	0	0	1	8
Lin et al., 2016	1	0	1	1	1	1	1	1	1	0	0	0	0	1	9
Liu et al., 2015	1	1	1	1	0	0	0	1	1	0	0	0	1	1	8
Ljungman et al., 2000	1	1	0	1	1	0	0	1	1	1	1	0	0	1	9
Martins et al., 2018	1	1	1	1	1	1	1	1	1	0	0	0	0	1	10
McCaffrey, 2006	0	1	1	1	1	1	1	1	1	0	0	1	0	1	10
McCarthy et al., 2016	1	1	1	1	1	0	1	1	1	0	0	1	0	1	10
Nazari et al., 2017	1	1	1	1	1	1	0	1	1	0	0	0	0	1	9
Rae et al., 2019	1	1	1	1	0	1	1	1	1	0	0	1	1	1	11
Reeve et al., 2020	1	1	1	1	1	1	1	1	1	1	1	1	0	1	13
Rosenberg et al., 2018	1	1	1	1	0	1	1	1	1	0	0	1	0	1	10
Sato et al., 2010	1	1	1	1	0	0	1	1	0	0	0	0	1	1	8
Scarpelli et al., 2008	1	1	1	1	1	1	1	1	1	0	0	1	0	1	11
Sitaresmi et al., 2008	1	1	1	1	0	0	0	1	1	0	0	0	0	1	7
Wu et al., 2016	1	1	1	1	0	0	0	1	1	0	0	0	1	1	8

**Table 2 ijerph-18-01911-t002:** Characteristics and psychometric properties of selected studies and reported instruments.

Self-Reported Anxiety Instrument	Study	Study Aims	QUADAS Score	Response Format	Validated Ages	Psychometric Properties Reported
	County			Pictorial Support		
Single-item instruments						
100-mm Visual Analogue scale (VAS)	Ljungman et al., 2000 [26]; Sweden	To test whether intranasal spray administration of midazolam could reduce anxiety, discomfort, pain, and procedure problems if given before insertion of a needle in a subcutaneously implanted central venous port. Furthermore, tolerability and side effects were investigated.	9 (64%)	Questionnaire; pictorial support	7–18 years (*n* = 30)	Validity Significant differences in fear/anxiety of having the needle inserted in the IV port with age (*p* = 0.024), in which the younger children were more anxious than the older children
Multi-item instruments						
Hospital Anxiety and Depression Scale—Anxiety (HADS-A)	Hedström et al., 2005 [27]; Sweden	To investigate perceptions of distress among adolescents recently diagnosed with cancer.	10 (71%)	4-point Likert scale and questionnaire; no pictorial support	13–15 years: *n* = 35 16–19 years: *n* = 21	Reliability α = 0.66.
	Jörngården et al., 2007 [28]; Sweden	To add to knowledge about HRQL, anxiety and depression among by following over time a group of individuals who probably will recover from their illness.	8 (57%)		13–15 years *n* = 35 16–19 years *n* = 21	Validity The mean difference between T1 and T4 was 1.74 (*t* = 2.62, *p* < 0.05)
	Lin et al., 2016 [29]; Taiwan	To evaluate nurse-led management model of adolescents acute lymphoblastic leukemia patients and improve their psychological care and quality of life.	9 (64%)		13 years (*n* = 73)	Validity A significant difference in HADS-A scores as a function of time between the groups (*p* < 0.05)
	Rae et al., 2019; [24]; Canada	To determine cut off points for newly developed Cancer Distress Scale (CDS)-a new patient.	11 (79%)		15 to 39 years (*n* = 515/*n* = 453; 15 to 19 years 25%)	Reliability HADS-A, a sensitivity of 0.78 and specificity of 0.79
	Rosenberg et al., 2018 [30]; USA	To determine whether Promoting Resilience in Stress Management (PRISM) improved psychosocial Outcomes in comparison with psychosocial usual care (UC).	10 (71%)		12–17 years, *n* = 67 18–25 years, *n* = 25	Validity HADS-A (>7), Usual care *n* = 14 and PRISM *n* = 11, non-significant difference (*p* = 0.24)
Kessler Psychological Distress Scale (K10)	McCarthy et al., 2016 [25]; Australia	To investigate the prevalence and predictors of psychological distress in adolescent and young adult (AYA) cancer patients and their parent caregivers.	10 (71%)	5-point Likert scale and questionnaire; no pictorial support	15 to 25 years (*n* = 196)	Reliability α = 0.93
Patient-Reported Outcomes Measurement Information System (PROMIS) and Chinese version of Pediatric Patient-Reported Outcomes Measurement Information System (C-Ped PROMIS)	Liu et al., 2015 [31]; China	To examine the measurement properties, e.g., scale dimensionality, item local dependence, and differential item functioning (DIF), of the C-Ped-PROMIS Anxiety and Depression short form measures by analyzing the emotional distress of children and adolescents with cancer in China.	8 (57%)	5-point Likert scale; no pictorial support	8 to 17 years (*n* = 232)	Reliability
						Reliability of 0.70
						Validity Factor analysis 0.59 to 0.82 for Anxiety, correlation coefficient of 0.76 (*p* < 0.001)
	Reeve et al., 2020 [32]; Canada	To evaluate the construct validity of the PROMIS Pediatric measures in a much larger sample than previous studies of children and adolescents undergoing active cancer treatment.	13 (93%)		7 to 18 years (*n* = 482)	Validity PROMIS Pediatric Psychological Stress measure was highly associated with anxiety (r = 0.75)
PedsQL^TM^ 3.0 Brain Tumor module	Sato et al., 2010 [33]; Japan	To investigate the feasibility, reliability, and validity of the Japanese version of the pediatric quality of life Brain Tumor module.	8 (57%)	5-point Likert scale and questionnaire; pictorial support: faces scale for 5- to 7-year-olds	5 to 18 years (*n* = 137)	Reliability α = 0.82 was obtained for the child self-reports on the procedural anxiety scale. α = 0.75, 0.85, and 0.85 were obtained for young children (5- to 7-year-olds), children (8- to 12-year-olds), and adolescents (13- to 18-year-olds), respectively
PedsQL^TM^ 3.0 Cancer Module	Germann et al., 2015 [34]; USA	To determine the pattern of resilience and adjustment (as measured by hope, anxiety, depression, and QoL) over the first year following cancer diagnosis, as well as (2) the longitudinal relationships between these variables, to inform future interventions toward hope in pediatric oncology patient.	11(79%)	5-point Likert scale and questionnaire; pictorial support: faces scale for 5- to 7-year-olds	8 to 17 years (*n* = 61)	Reliability α = 80 for the standardized scale in the current study
	Scarpelli et al., 2008 [35]; Brazil	To test the psychometric properties of the PedsQL cancer module scale cross-culturally adapted for Brazilian Portuguese.	11 (79%)		5 to 18 years (*n* = 124)	Reliability Version designed for children/adolescents (α = 0.76) Procedural anxiety subscale presented values near to or above α = 0.70 in all age groups
	Sitaresmi et al., 2008 [36]; Indonesia	To assess health related quality of life (HRQOL) in childhood acute lymphoblastic leukemia (ALL) patients in Indonesia and to assess the influence of demographic and medical characteristics of HRQOL.	7 (50%)		5 to 18 years (*n* = 55)	Reliability Procedural anxiety α = 0.85 Treatment anxiety α = 0.78
						Validity Procedural anxiety ICC = 0.61 Treatment anxiety ICC = 0.33
Revised Child Manifest Anxiety Scale (RCMAS/ RCMAS-2)	Martins et al., 2018 [37]; Portugal	To examine associations among self-reported hope, anxiety, and HRQoL in two clinical groups (on-treatment vs. off treatment) of children/adolescents with cancer	10 (71%)	Yes/no questions; no pictorial support	8 to 19 years (*n* = 211)	Reliability Scale scores α = 0.63 and α = 0.64 for children/adolescents on-treatment and off-treatment, respectively
	McCaffrey 2006 [38]; Australia	To determine the effectiveness of The Modified Feeling Great Program (MFGP) designed to reduce anxiety and boost self-concept.	10 (71%)		6 to 17 years (*n* = 20)	Reliability RCMAS-α = 0.87, test of self-concept 2 α = 0.92
	Nazari et al., 2017 [39]; Iran	To compare the quality of life, anxiety and depression in children with cancer and healthy children in Kermanshah, Iran.	9 (64%)		10 to 16 years (*n* = 60)	Validity A significant difference (*p* < 0.001) between children with and without cancer
	Wu et al., 2016 [40]; Taiwan	To translate the RCMAS-2 into Chinese and evaluate its psychometric properties in pediatric cancer patients in Taiwan.	8 (57%)		6 to 19 years (*n* = 370)	Reliability The internal consistency for the Total Anxiety score was 0.90 which indicates that the Chinese version of RCMAS-2 has good internal consistency: α = 0.65 for Physiological Anxiety and 0.77 for Social Anxiety
State Trait Anxiety Inventory (STAI) Trait and State scale	Allen et al., 1997 [41]; United Kingdom	To present the findings at the time of first diagnosis of a longitudinal study of the emotional impact of the diagnosis of cancer in patients and their families presenting to an adolescent cancer unit and of the coping strategies they employ.	10 (71%)	4-point Likert scale; no pictorial support	12 to 20 years (*n* = 43 + 173)	Reliability State anxiety α = 0.89; trait anxiety α = 0.86
	Burgess and Haaga, 1998 [42]; USA	To examine individual differences in emotional responses to cancer by applying Lazarus’s and Weiner’s cognitive models of emotion.	7 (50%)		12 to 18 years (*n* = 72)	Validity
						Trait anxiety (r = 0.51‚ *p* < 0.001) correlated positively with the CBCL Anxiety-Depression subscale
	Germann et al., 2015 [34]; USA	To determine the pattern of resilience and adjustment (as measured by hope, anxiety, depression, and QoL) over the first year following cancer diagnosis, as well as (2) the longitudinal relationships between these variables, to inform future interventions toward hope in pediatric oncology patient	11 (79%)		8 to 17 years (*n* = 61)	Reliability α = 0.81 for the State scale
STAI for Children (STAIC)	Sato et al., 2010 [33]; Japan	To investigate the feasibility, reliability, and validity of the Japanese version of the pediatric quality of life Brain Tumor module	8 (57%)	3-point Likert scale; no pictorial support	8 to 18 years (*n* = 106)	Reliability Internal consistencies for the State and Trait Anxiety scales were 0.89 and 0.89, respectively

**Table 3 ijerph-18-01911-t003:** Summary of the study characteristics.

Characteristics	Studies (*n* = 19)
Date of publication	
>2010	10 (53%)
≤2010	9 (47%)
Study location	
USA	3 (16%)
Sweden	3 (16%)
Australia	2 (11%)
Canada	2 (11%)
Taiwan	2 (11%)
Brazil	1 (5%)
China	1 (5%)
Indonesia	1 (5%)
Iran	1 (5%)
Japan	1 (5%)
Portugal	1 (5%)
United Kingdom	1 (5%)
Pictorial support	3 (16%)
Instrument, *n* (%)	(*n* = 19) *
Single-item instruments	
Visual Analogue Scale (VAS) **	1 (5%)
Multi-item instruments	
Hospital Anxiety and Depression Scale (HADS)	5 (26%)
Kessler Psychological Distress Scale (K10)	1 (5%)
Patient-Reported Outcomes Measurement Information System (PROMIS)	2 (11%)
Pediatric quality of life (PedsQL™) 3.0 Brain Tumor module **	1 (5%)
Pediatric quality of life (PedsQL™) 3.0 Cancer Module **	3 (16%)
Revised Children’s Manifest Anxiety Scale (RCMAS/RCMAS-2)	4 (21%)
State-Trait Anxiety Inventory (STAI-A/STAI-T); State-Trait Anxiety Inventory for Children (STAIC)	4 (21%)

* Two instruments (PedsQL™ 3.0 Brain Tumor; STAIC) are represented in one study [33]; two instruments (PedsQL™ 3.0 Cancer Module; STAI) are represented in one study [34]. ** Instrument with pictorial support.

## Data Availability

The datasets used and/or analysed during the current study is available from the corresponding author on reasonable request.

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
