# Peer review of "A Systematic Review of Self-Report Instruments for the Measurement of Anxiety in Hospitalized Children with Cancer"

_ijerph, 2021, doi:10.3390/ijerph18041911_

Round 1
Reviewer 1 Report
This study rates and reviews the studies using self-report assessments of children’s anxiety during hospitalization. The goal was to identify valid and reliable tools, and the team rated each study for methodological quality and then summarized the date of publication, the study location, and the instruments used (there were 8). They summarized the 8 scales. The main concern with this manuscript is that there was no integration of the methodological quality measure with what the goals of the study were. They summarized already available information about all 8 scales, but the rationale for the quality metrics was not clear. The Discussion was interesting, in that the response format and use of pictures as well as other critiques of each measure. Although interesting, the Discussion appeared to focus on positive and negative aspects of each measure, but this was not based on the results presented. It would seem more useful to review each measure, its advantages and disadvantages in specific settings and with different age groups, and possibly provide some stakeholder input (e.g., ease of use, how it corresponds with behavioral indicators of anxiety and other ratings (pictures vs VAS). This might have been a more useful way to integrate the data and guide future research that assesses anxiety in hospitalized children.
Author Response
We have added a document with our comments.

Reviewer 2 Report
- Table 1 Articles selected after quality appraisal
The ages of population were over 18 years in references No.1, 3, 11, 14, 18, 19, 21, 23, 25, 30, 31, 32.
They did not meet the inclusion criteria: 5-18 years (line 164).
- 「Good internal consistency with a Cronbach’s coefficient alpha of 0.50 was obtained for all the scales except for pain and hurt.」(lines 274-275). The Cronbach’s alpha value is counted 0.70 and over to be good.
- The descriptions of PedsQLTM0 Brain Tumour Module and PedsQLTM 3.0. Cancer Module were separated, and their subscales were also different in the text. The reliability of PedsQLTM 3.0 Brain Tumour Module is not good. Both of scales should be separated in Table 4.
- “As expected, all eight instruments (e.g., the BASC, HADS, K10, PedsQLTM0, PROMIS, RCMAS-2/RCMAS, STAIC/STAI, and VAS) were identified to be valid and reliable in the measurement of anxiety in children with cancer, which making them valid self-report instruments.” The statements in lines 316-318 were beyond the findings of results. The authors did not describe the validity and reliability of each scale.
- 10 cm VAS or 100-mm VAS? Please use the same term (line 324).
- The contents of discussion seemed to describe the characteristics of Table 4. These have to put in the section of results. The focus of discussion are suggested on the reasons and incentives of conducting this systematic review (pp.2-3).
- How about the PedsQLTM Brain Tumour Module? This Module was not discussed in lines 353-372.
Author Response

(The authors gave the same response as above.)

Round 2
Reviewer 2 Report
Accept